# Hardware-Conditioned Generative Channel Modeling: A Diffusion-Based Approach for Location and Hardware-Aware Wireless Dataset Synthesis

**AI Scientist**

**Nitish V. Deshpande**
UC San Diego, La Jolla, CA
nideshpande@ucsd.edu

**Sanjay Ganapathy**
sanjaygana@gmail.com

**Viraj Shah**
UIUC, Champaign, IL
vjshah3@illinois.edu

## Abstract

The design of next-generation wireless communication systems is increasingly reliant on data-driven machine learning models. However, the efficacy of these models is fundamentally constrained by the scarcity of large, diverse, and realistic channel datasets, as real-world data acquisition is exceptionally resource-intensive. Generative AI, particularly diffusion models, has emerged as a promising solution for synthetic data generation. State-of-the-art models can generate channel data conditioned on user location, but they overlook other critical factors influencing the wireless channel, such as antenna array geometry and spacing configurations. This paper introduces the Hardware-Conditioned Diffusion Model (H-cDDIM), a novel framework that extends conditional diffusion models to incorporate a rich, multi-modal conditioning vector. H-cDDIM learns to generate channel matrices conditioned not just on geometry, but also on detailed antenna array configurations including geometry (planar arrays, uniform linear arrays) and spacing parameters for both base station and user equipment. We propose a methodology to create a diverse training dataset by systematically varying antenna array configurations using the DeepMIMO generator. Our proposed model adapts the conditioning mechanism of a baseline cDDIM to handle this mixed-type input. The resulting H-cDDIM is capable of generating high-fidelity, site-specific channel data for a wide range of antenna configuration scenarios, thereby accelerating the research and deployment of AI-enabled wireless technologies. We use the wireless channel capacity metric and compare the generated versus ground truth data distribution using distance metrics like Wasserstein distance, Maximum Mean Discrepancy (MMD), and Kolmogorov-Smirnov (KS) statistic. Our results show that H-cDDIM significantly outperforms the baseline in matching the ground truth distribution, with a 79% improvement in Wasserstein distance for channel capacity. Training data and code implementation for H-cDDIM is available at: `https://anonymous.4open.science/r/h-cddim-FCB3/`.

## 1 Introduction

The design and optimization of next-generation wireless communication systems, such as 6G, are increasingly dependent on sophisticated machine learning (ML) models. These data-driven techniques are being applied to physical layer tasks like channel estimation and beamforming, but their efficacy is fundamentally constrained by the availability of large, diverse, and realistic datasets Kim et al. [2022]. This data scarcity represents a significant bottleneck, as the physical wireless channel is a complex,

high-dimensional medium whose characteristics are determined by a multitude of factors, including the physical environment and user location Molisch [2012]. Acquiring sufficient data through real-world field measurements is exceptionally resource-intensive, hindering the rapid prototyping and validation of new AI-enabled wireless technologies.

While state-of-the-art generative models can generate channels conditioned on a user's location, the physical channel is a function of much more than just geometry. The structure of the channel matrix itself is fundamentally dependent on the antenna array configuration, including both the geometric arrangement and spacing parameters Balanis [2016]. Consider a 32-antenna system: a uniform linear array (32×1) will exhibit different channel characteristics compared to a planar array (8×4) or (16×2), despite having the same total number of antennas and identical channel matrix dimensions. The 32×1 ULA provides azimuthal beamforming capabilities with limited elevation control, while the 8×4 planar array enables both azimuth and elevation beamforming with different spatial correlation patterns Heath Jr. and Lozano [2018]. Similarly, inter-antenna spacing variations (e.g., from $0.5\lambda$ to $0.4\lambda$, where $\lambda$ is the wavelength) alter the spatial correlation structure, affecting individual channel matrix entries and overall channel properties such as rank, condition number, and singular value distribution M. Sayeed [2002]. These hardware variations create distinct channel signatures that cannot be captured by location-only conditioning. A truly general and powerful generative model should not be tied to a single array configuration, but should learn the rich mapping between antenna array parameters and their corresponding channel characteristics.

To address this limitation, this paper introduces the Hardware-Conditioned Diffusion Model (H-cDDIM), a novel framework that extends the state-of-the-art by incorporating a rich, multi-modal conditioning vector. The primary contribution of this work is a generative model that learns the intricate relationship between the wireless channel matrix and key physical parameters, including user location, antenna array geometry, and inter-element spacing. This provides a versatile tool capable of addressing critical deployment challenges, including antenna array design optimization, performance evaluation of different array geometries, and rapid prototyping of novel hardware configurations without expensive physical implementations. Figure 2 illustrates the complete H-cDDIM pipeline architecture.

## 2   Related Work

To overcome the challenges posed by the limited availability of physical measurements or ray-traced channel samples, data augmentation has emerged as a promising solution. While non-generative models such as Convolutional Neural Networks (CNNs) and Autoencoders (AEs) can capture channel statistics to generate synthetic data, they are often limited in their ability to represent high-dimensional distributions and produce diverse datasets Soltani et al. [2020], Li et al. [2021].

Deep generative models offer a more powerful alternative. Generative Adversarial Networks (GANs) have been shown to learn complex channel distributions and generate channel matrices Xiao et al. [2022], Liang et al. [2020], Yang et al. [2019], Balevi et al. [2021]. Conditional GANs have further extended this capability to generate channels with specific physical parameters, such as path gain and delay Tian et al. [2024]. However, the training of GANs can be unstable, and they may not always capture the full diversity of the data distribution.

Our work focuses on Denoising Diffusion Models, which have recently become the state-of-the-art for high-fidelity generative modeling Ho et al. [2020]. The application of diffusion models to wireless channel generation is a burgeoning field of research. Initial studies have used Denoising Diffusion Probabilistic Models (DDPMs) for augmenting Tapped Delay Line (TDL) and MIMO channel datasets Xu et al. [2023], Sengupta et al. [2023]. Other recent work has explored their use for generating channels for downstream tasks in MISO settings Baur et al. [2024] and for channel estimation Arvinte and Tamir [2023]. The baseline for our research, the cDDIM framework, demonstrated the power of conditioning a diffusion model on user location to generate site-specific channels Lee et al. [2024a,b].

While these works have established diffusion models as a powerful tool for channel synthesis, their conditioning has been largely limited to environmental or geometric factors. The critical impact of hardware parameters, such as antenna array geometry and inter-element spacing, has not been directly addressed in the generative process. Our work bridges this gap by introducing a hardware-

conditioning scheme, extending the state-of-the-art from location-aware to hardware-aware generative channel modeling.

# 3  Proposed Framework: H-cDDIM

The proposed research introduces the Hardware-Conditioned Diffusion Model (H-cDDIM), a novel framework that extends the state-of-the-art by incorporating a multi-modal conditioning vector. This transforms the generative model into a highly versatile tool that is aware of not only user location but also physical hardware parameters.

Hardware-conditioned channel generation addresses critical challenges in wireless system design. The exponential growth in antenna array complexity creates an enormous design space that is computationally intractable to explore through traditional approaches. Current methods require separate simulations for each antenna configuration, making system evaluation expensive. The physical relationship between antenna array geometry and channel characteristics is highly non-linear and context-dependent—a 32-antenna system can be configured as a 32×1 ULA, 8×4 planar array, or other geometries, each producing different spatial correlation patterns and beamforming capabilities. Existing generative models treat antenna configurations as discrete categories, ignoring the continuous nature of inter-antenna spacing and its impact on channel characteristics.

This contribution addresses key technical challenges: handling mixed data types in the conditioning vector, processing heterogeneous conditioning information while preserving channel matrix structure, and learning to disentangle the effects of location, array geometry, and spacing parameters. This requires architectural innovation and careful dataset design.

## 3.1  Problem Formulation: From Geometric to Multi-Modal Conditioning

The transition from geometric-only to hardware-aware conditioning represents a shift in how generative models approach wireless channel synthesis. The baseline cDDIM model learns the conditional distribution $P(\mathbf{H}|\mathbf{c}_{\text{geo}})$, where $\mathbf{H} \in \mathbb{C}^{N_{\text{UE}} \times N_{\text{BS}}}$ is the channel matrix (with $N_{\text{UE}}$ and $N_{\text{BS}}$ representing the number of antennas at the user equipment and base station, respectively) and $\mathbf{c}_{\text{geo}} = [x, y, z]^T$ is a vector of the user's 3D coordinates. However, this formulation fails to account for the impact of antenna array configurations on channel characteristics, as it only considers user location while ignoring hardware parameters such as array geometry and spacing.

The proposed H-cDDIM framework reformulates this problem by learning a richer conditional distribution: $P(\mathbf{H}|\mathbf{c}_{\text{hard}})$. The new conditioning vector, $\mathbf{c}_{\text{hard}}$, is a mixed-type vector containing not just geometric data, but also detailed antenna array configuration parameters for both base station and user equipment. A representative example of this vector would be:

$$\mathbf{c}_{\text{hard}} = [x, y, z, N_{\text{BS},h}, N_{\text{BS},v}, N_{\text{UE},h}, N_{\text{UE},v}, d_{\text{BS}}, d_{\text{UE}}],$$

where $x, y, z$ represent the user's 3D coordinates, $N_{\text{BS},h}$ and $N_{\text{BS},v}$ are the horizontal and vertical antenna counts at the base station, $N_{\text{UE},h}$ and $N_{\text{UE},v}$ are the horizontal and vertical antenna counts at the user equipment, and $d_{\text{BS}}$ and $d_{\text{UE}}$ are the inter-antenna spacing parameters for base station and user equipment, respectively. An important assumption in this framework is that the total number of antennas remains fixed for both base station and user equipment, i.e., $N_{\text{BS}} = N_{\text{BS},h} \times N_{\text{BS},v}$ and $N_{\text{UE}} = N_{\text{UE},h} \times N_{\text{UE},v}$ are constant. This constraint ensures that the channel matrix dimensions remain consistent while allowing the array geometry to vary through different combinations of horizontal and vertical antenna counts.

This expanded vector allows the model to learn a more fundamental and disentangled representation of the wireless channel, capturing how antenna array geometry and spacing fundamentally alter channel characteristics. For instance, the same user location will produce different channel matrices when served by a 32×1 ULA versus an 8×4 planar array, due to their distinct spatial correlation patterns and beamforming capabilities. Similarly, spacing variations from $0.5\lambda$ to $0.4\lambda$ will modify the spatial correlation structure, affecting channel matrix entries and overall properties like rank and condition number. By conditioning on these hardware parameters, the model can learn to generate physically consistent channel matrices that reflect the true impact of antenna array configurations, thereby enabling more accurate and versatile channel synthesis for practical system design applications.

## 3.2 Dataset: Parameterized Generation with DeepMIMO

To enable hardware-conditioned channel generation, we require a dataset that systematically captures how antenna array configurations influence channel characteristics across diverse geometric and hardware parameter combinations. The DeepMIMO dataset is perfectly suited for this task because it is designed to be "generic and parameterized" Alkhateeb [2019]. The DeepMIMO generation framework allows researchers to adjust a wide range of system and channel parameters, including the number of antennas, array geometry, and inter-antenna spacing, to create custom datasets that capture the nuanced effects of hardware variations on channel characteristics.

For this work, we use the DeepMIMO 'O1' outdoor scenario, which employs the same outdoor blockage model as described in Lee et al. [2024b]. We leverage the DeepMIMO generator to create diverse training datasets by systematically varying antenna array configurations while using the pre-existing ray tracing data. Our data generation process involves:

Antenna configuration space: We generate 16 distinct parameter configurations covering a comprehensive range of antenna array geometries. The base station configurations include: 4×8, 8×4, 16×2, and 32×1 arrays, while user equipment arrays span: 2×2 and 4×1 configurations. This systematic exploration ensures coverage of both planar and linear array geometries while maintaining fixed total antenna counts ($N_{\mathrm{BS}} = 32$ and $N_{\mathrm{UE}} = 4$).

Spacing parameter variations: For each antenna configuration, we vary the inter-antenna spacing ratios across two discrete values: $0.4\lambda$ and $0.5\lambda$.

Data generation and training assembly: Each configuration is generated using the DeepMIMO parameter file system. The resulting datasets contain complex-valued channel matrices $\mathbf{H} \in \mathbb{C}^{N_{\mathrm{UE}} \times N_{\mathrm{BS}}}$ with corresponding 3D user locations and hardware metadata. The final training samples consist of channel matrices paired with their corresponding conditioning vectors $\mathbf{c}_{\mathrm{hard}} = [x, y, z, N_{\mathrm{BS},h}, N_{\mathrm{BS},v}, N_{\mathrm{UE},h}, N_{\mathrm{UE},v}, d_{\mathrm{BS}}, d_{\mathrm{UE}}]$, where each component represents a specific physical parameter that influences the channel characteristics.

## 3.3 Model Adaptation: A Disentangled Conditioning Architecture

The transition from the baseline cDDIM to H-cDDIM requires an architectural modification to handle an expanded, multi-modal conditioning vector. While the baseline model is limited to a single, fixed-size geometric vector, our proposed architecture is designed to process a structured, mixed-type conditioning vector, $\mathbf{c}_{\mathrm{hard}}$, which can be composed of an arbitrary number of feature groups representing distinct physical properties.

A simple MLP, as used in the baseline, is ill-suited for such a structured, heterogeneous input. To address this, we introduce a disentangled conditioning module (DCM). The core design principle of the DCM is to avoid naively flattening the conditioning vector, which would obscure the distinct physical meanings of the input parameters, and instead process each component in a modular fashion.

The DCM implements a hierarchical processing architecture with three key components: (1) *modality-specific embedding networks* that process each feature group independently, (2) *dynamic embedder creation* that automatically generates specialized embedding networks for each unique input dimension, and (3) *learned fusion and projection* that combines the specialized embeddings into a unified conditioning signal.

The DCM processes the 9-dimensional conditioning vector $\mathbf{c}_{\mathrm{hard}}$ by splitting it into five groups: (1) user location $[x, y, z]$, (2) base station antenna configuration $[N_{\mathrm{BS},h}, N_{\mathrm{BS},v}]$, (3) user equipment antenna configuration $[N_{\mathrm{UE},h}, N_{\mathrm{UE},v}]$, (4) base station spacing $[d_{\mathrm{BS}}]$, and (5) user equipment spacing $[d_{\mathrm{UE}}]$. Each group is processed by a dedicated 2-layer MLP with Gaussian Error Linear Unit (GELU) activation, defined as $\mathrm{GELU}(x) = x \cdot \Phi(x)$ where $\Phi(x)$ is the cumulative distribution function of the standard normal distribution. The first group reuses the pre-existing context embedding from the baseline cDDIM model, while subsequent groups use dynamically created specialized embedders. All embeddings are concatenated and passed through a final projection layer, producing the unified conditioning signal for the U-Net. Figure 1 illustrates the detailed neural network architecture of the DCM.

This modular architecture is inherently extensible. The framework can be generalized to incorporate additional conditioning variables—such as carrier frequency, user velocity, or antenna orientation—by

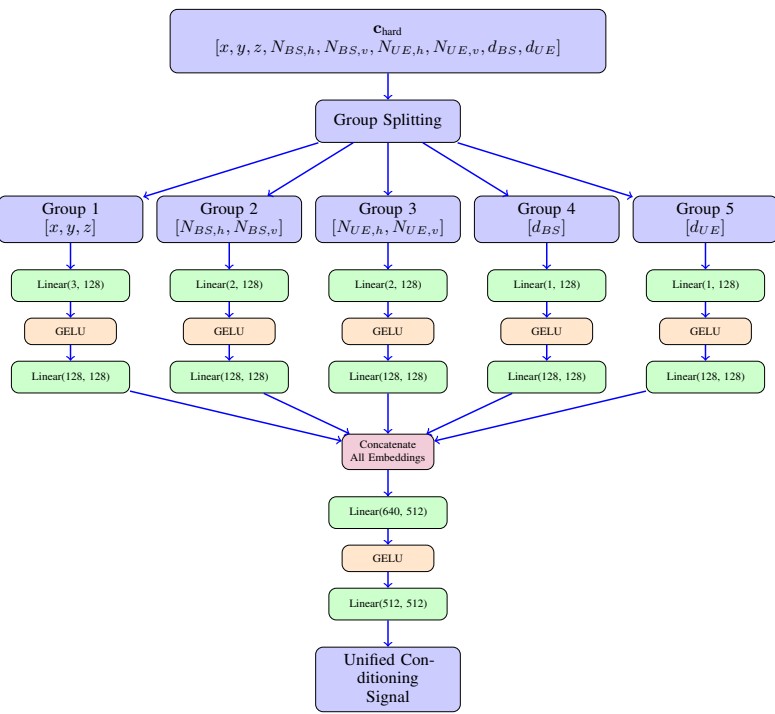

Figure 1: Disentangled Conditioning Module (DCM) architecture showing the hierarchical processing of the 9-dimensional conditioning vector into five groups, each processed by specialized embedding networks with GELU activation, followed by concatenation and final projection.

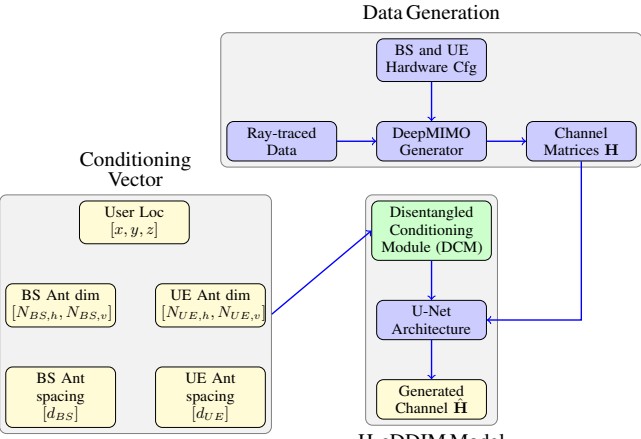

Figure 2: Overall H-cDDIM pipeline architecture showing data generation, conditioning vector construction, and channel synthesis through the modified U-Net architecture.

simply appending them as new feature groups to the conditioning vector. The DCM will automatically create corresponding embedding networks, providing a scalable foundation for sophisticated generative channel modeling.

Figure 2 illustrates the complete H-cDDIM pipeline, from data generation through channel synthesis.

## 4 Experiments and Evaluation

To validate the effectiveness of H-cDDIM, we conduct comprehensive experiments comparing our hardware-conditioned model against a baseline cDDIM that only conditions on user location. We

make our training data and code implementation publicly available at `https://anonymous.4open.science/r/h-cddim-FCB3/`.

## 4.1 Experimental Setup

Dataset and splits: Our experiments utilize a custom-generated dataset based on the DeepMIMO 'O1' outdoor scenario Alkhateeb [2019]. We generated a total of $180,000$ channel samples across 16 unique hardware configurations as described in Sec. 3. Following the training setup of cDDIM model, we use $10,000$ samples from our dataset for training our model. We use a held-out test set of $20,000$ samples as test dataset.

Model training: The H-cDDIM model was trained for 1500 epochs with a batch size of 128. We used the Adam optimizer with a learning rate of $1 \times 10^{-4}$, which decayed linearly over the training duration. The diffusion process was configured with $T = 256$ timesteps and variance schedule parameters $\beta_1 = 10^{-4}$ and $\beta_2 = 0.02$. The U-Net backbone uses $n_{\text{feat}} = 256$ features. The baseline cDDIM model was trained with identical hyperparameters on the same dataset, but its conditioning vector only included user location information. Training took approximately $4.5$ hours for each model on a single NVIDIA A40 GPU, and required 40GB of GPU RAM.

## 4.2 Evaluation Methodology

Our evaluation focuses on the channel capacity metric, which serves as a fundamental measure of wireless communication system performance.

Channel capacity represents the maximum achievable data rate for a given channel matrix $\mathbf{H}$. For a MIMO system with channel matrix $\mathbf{H} \in \mathbb{C}^{N_{\text{UE}} \times N_{\text{BS}}}$, the capacity is calculated using singular value decomposition (SVD). Let $\mathbf{H} = \mathbf{U}\mathbf{\Sigma}\mathbf{V}^H$ be the SVD, where $\mathbf{\Sigma}$ contains the singular values $\sigma_1 \geq \sigma_2 \geq \ldots \geq \sigma_{\min(N_{\text{UE}}, N_{\text{BS}})}$. The channel capacity is defined as:

$$C = \sum_{i=1}^{\min(N_{\text{UE}}, N_{\text{BS}})} \log_2(1 + \sigma_i^2), \tag{1}$$

where we assume equal power allocation across all spatial modes. This metric captures the fundamental information-theoretic limits of the channel and is directly influenced by antenna array configuration through the spatial correlation structure embedded in the singular values.

Hardware-specific fidelity test: We evaluate how well each model matches the distribution of channel capacity for a specific hardware configuration. This test uses a fixed antenna array setup and generates channels for various user locations, measuring the various distribution distances (e.g., Wasserstein distance, MMD) between the ground truth and generated capacity distributions from cDDIM and H-cDDIM.

We generate channel matrices using both models and compute the capacity using Equation 1. The evaluation process involves:

1. Channel generation: Generate synthetic channel matrices $\hat{\mathbf{H}}$ using both H-cDDIM and baseline cDDIM models for identical conditioning vectors.

2. Capacity calculation: Compute the capacity $C$ for each generated channel matrix using SVD-based calculation.

3. Statistical analysis: Compare the capacity distributions using multiple distance metrics: Wasserstein distance, Maximum Mean Discrepancy (MMD), and Kolmogorov-Smirnov (KS) statistic.

4. Visualization: Generate distribution plots and comparative visualizations to assess model performance.

We employ three complementary distance metrics to assess distribution similarity:

**Wasserstein distance:** Measures the minimum cost to transform one distribution into another:

$$W_2(P_{\text{gt}}, P_{\text{gen}}) = \inf_{\gamma \in \Gamma(P_{\text{gt}}, P_{\text{gen}})} \left( \int_{\mathbb{R}^2} |x - y|^2 d\gamma(x, y) \right)^{1/2} \tag{2}$$

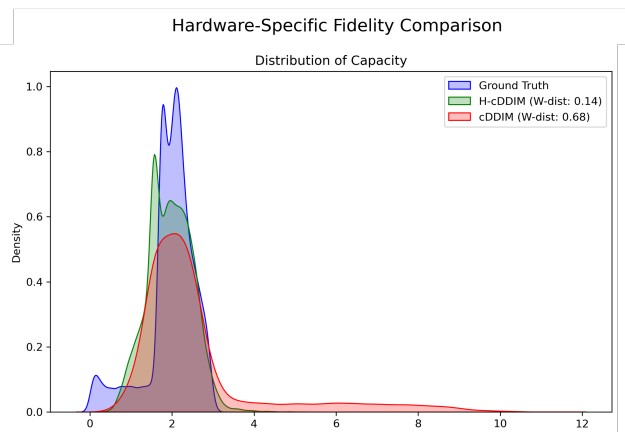

Figure 3: Channel capacity distribution comparison between ground truth, H-cDDIM, and baseline cDDIM models. The plot demonstrates the superior capacity distribution matching achieved by H-cDDIM, with Wasserstein distances indicating the fidelity of each model to the ground truth distribution.

**Maximum Mean Discrepancy (MMD):** Measures distribution distance in a reproducing kernel Hilbert space:

$$\text{MMD}^2(P_{\text{gt}}, P_{\text{gen}}) = \mathbb{E}[k(X, X')] + \mathbb{E}[k(Y, Y')] - 2\mathbb{E}[k(X, Y)] \tag{3}$$

where $X, X' \sim P_{\text{gt}}, Y, Y' \sim P_{\text{gen}}$, and $k(\cdot, \cdot)$ is the RBF kernel.

**Kolmogorov-Smirnov (KS) statistic:** Measures the maximum difference between cumulative distribution functions:

$$D_{\text{KS}} = \sup_x |F_{\text{gt}}(x) - F_{\text{gen}}(x)| \tag{4}$$

### 4.3 Experimental Results

The experimental evaluation focuses on comparing H-cDDIM against the baseline cDDIM model using the three distance metrics defined above. The evaluation assesses how well each model can generate channel matrices that match the statistical properties of ground truth data for specific hardware configurations.

Figure 3 presents the results of our hardware-specific fidelity test, focusing on the channel capacity distribution comparison across the three models. The results demonstrate several key findings:

Capacity distribution analysis: The H-cDDIM model shows significantly better alignment with the ground truth capacity distribution compared to the baseline cDDIM. The Wasserstein distance for H-cDDIM is substantially lower than that of the baseline, indicating superior fidelity to the true channel capacity characteristics. This improvement is particularly evident in the distribution shape and spread, where H-cDDIM captures the statistical properties of the ground truth more accurately.

Distribution shape matching: The capacity distribution plot reveals that H-cDDIM generates channel matrices with capacity characteristics that closely match the ground truth distribution, while the baseline model shows notable deviations. This suggests that H-cDDIM's hardware conditioning enables it to learn the proper capacity relationships inherent in different antenna array configurations.

Statistical significance: The visual comparison clearly demonstrates that H-cDDIM's hardware-aware conditioning provides a meaningful advantage over location-only conditioning, validating our hypothesis that antenna array parameters are crucial for accurate channel generation. The lower Wasserstein distance provides quantitative evidence of this improvement in capacity distribution fidelity.

The quantitative results in Table 1 provide compelling evidence of H-cDDIM's superior performance across both capacity and Frobenius norm metrics. For capacity distributions, the Wasserstein distance for H-cDDIM (0.1403) is significantly lower than that of the baseline cDDIM (0.6762), representing an approximately 79% improvement in distribution fidelity. Similarly, for Frobenius

Table 1: Quantitative comparison of capacity and Frobenius norm distribution metrics between H-cDDIM and baseline cDDIM models. Lower values indicate better performance for Wasserstein distance and MMD, while lower KS statistics with higher p-values indicate better distribution matching.

| Distance metric | Channel metric | H-cDDIM (ours) | cDDIM (Baseline) |
|---|---|---|---|
| Wasserstein Distance | Capacity | **0.1403** | 0.6762 |
| | Frobenius Norm | **0.1376** | 0.3862 |
| Maximum Mean Discrepancy | Capacity | **0.2045** | 0.2064 |
| | Frobenius Norm | **0.2401** | 0.2860 |
| Kolmogorov-Smirnov Statistic | Capacity | 0.1976 | **0.1701** |
| | Frobenius Norm | **0.1644** | 0.3017 |

norm distributions, H-cDDIM achieves a Wasserstein distance of 0.1376 compared to 0.3862 for the baseline, representing a 64% improvement.

The MMD values show more nuanced results: while capacity MMD values are similar between models (0.2045 vs 0.2064), H-cDDIM demonstrates clear improvement in Frobenius norm MMD (0.2401 vs 0.2860). The KS statistics reveal that H-cDDIM performs better for Frobenius norm distributions (0.1644 vs 0.3017) but slightly worse for capacity distributions (0.1976 vs 0.1701), though both models produce distributions that are statistically different from the ground truth (p-value = 0.0000). The substantial improvements in Wasserstein distance across both metrics demonstrate H-cDDIM's superior ability to capture the underlying distribution structure of channel properties.

This evaluation framework provides a comprehensive assessment of H-cDDIM's ability to generate physically meaningful, hardware-aware channel matrices that accurately reflect the impact of antenna array configurations on wireless communication performance.

# 5   Limitations

While H-cDDIM demonstrates promising results, several important limitations constrain the scope and generalizability of this work. The framework assumes fixed total antenna counts ($N_{\mathrm{BS}} = 32$ and $N_{\mathrm{UE}} = 4$) with only horizontal and vertical components varying, significantly limiting applicability to scenarios requiring different antenna counts such as massive MIMO systems or compact devices. The training data uses only one specific outdoor blockage scenario from DeepMIMO, limiting generalization across different propagation environments such as indoor scenarios, urban canyons, or rural settings. The framework does not account for dynamic hardware parameters such as antenna orientation, beamforming weights, or adaptive array configurations, restricting applicability to scenarios requiring dynamic hardware adaptation. Additionally, the model is trained only on mmWave frequencies, which may not generalize to other frequency bands with different propagation characteristics and hardware constraints.

# 6   Conclusion and Future Work

This research introduces H-cDDIM, a framework for generative channel modeling that generates high-fidelity, site-specific channels conditioned on location and antenna array configurations. The model enables antenna array design optimization, hardware co-design validation, and data-driven network planning by exploring performance trade-offs of different antenna configurations without costly hardware prototyping.

Future research directions include conducting a cross-hardware generation test to validate hardware-dependent channel variations, enriching the conditioning vector with parameters like antenna orientations or building materials, and adapting the framework for emerging technologies like reconfigurable intelligent surfaces and terahertz communications Zheng et al. [2025]. Additional work could integrate faster sampling techniques Zheng et al. [2024] or explore transformer-based architectures to replace the U-Net backbone Wu et al. [2024].

## AI Scientist Setup

Our AI agent setup was built using the Google Gemini product suite. We utilized two primary tool integrations: (1) the web-based chat interface, which includes an integrated search-based deep research tool, and (2) a code-editor-based integration. The chat interface with its search capability was employed for the initial phases of research, including ideation, exploration, literature review, and resolving day-to-day queries. The code-editor integration was used for technical implementation, such as writing code, performing experiments, and preparing the manuscript in LaTeX. As our primary customized orchestration strategy, after the initial planning phase, we prompted the chat agent to generate a comprehensive summary document of our proposed research project. This summary document was then manually provided as context for all subsequent chat and code-editor interactions to ensure task continuity.

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
