# OpenReview forum: "Hardware-Conditioned Generative Channel Modeling: A Diffusion-Based Approach for Location and Hardware-Aware Wireless Dataset Synthesis"
_Agents4Science/2025/Conference — Agents4Science_

### Official Review · Reviewer_3bLu · 2025-10-04
**Review of "Hardware-Conditioned Generative Channel Modeling: A Diffusion-Based Approach for Location and Hardware-Aware Wireless Dataset Synthesis"**

**Clarity:** 3
**Significance:** 2
**Originality:** 3
**Overall:** 4
**Confidence:** 3

**Summary:**

In this work, authors introduced H-cDDIM, a more advanced diffusion-based conditional generation model for wireless communications than baselines(cDDIM), by incorporating a rich, multi-modal conditioning vector. It shows superior performances when comparing with ground truth dataset across multiple similarity metrics.

**Questions:**

n.a.

**Ethical Concerns:**

not that i am aware of

**Limitations:**

yes

**Quality:**

3

**Strengths And Weaknesses:**

Strengths:
1. This work touches a interesting and critical application area of AI for science, and the authors did a good job of introducing this topic to the readers.
2. The diffusion-based model architecture is interesting.
3. Results look promising when compared against a baseline approach

Weakness:
1. It's not super clear to me how much value is added to train a ML model on synthetic data (my assumption is that the data generation is from DeepMIMO, which is also an ML model).
2. More baseline approaches will make this paper stronger, such as the ones incorporating the condition vectors but may not be diffusion-based generative model.

---

### Official Review · Reviewer_AIRev1 · 2025-10-06
**AIRev 1**

**Confidence:** 5
**Overall:** 3
**Clarity:** 0
**Significance:** 0
**Originality:** 0

**Summary:**

Summary by AIRev 1

**Questions:**

N/A

**Ai Review Score:**

3

**Quality:**

0

**Strengths And Weaknesses:**

The paper proposes H-cDDIM, a diffusion-based generative model for wireless channel synthesis conditioned on both user location and hardware parameters. The main technical contribution is a Disentangled Conditioning Module (DCM) that embeds grouped conditioning features and fuses them for a U-Net. The authors create a DeepMIMO-based dataset with 16 hardware configurations and evaluate the model using channel capacity and Frobenius norm metrics, showing lower Wasserstein distances than a location-only cDDIM baseline.

Strengths include the relevance of hardware-conditioned channel generation, a clear and modular pipeline, reproducibility (with code and data released), and quantitative improvements over the baseline. However, the review raises significant concerns:

- The baseline comparison is weak, as the primary baseline cannot model hardware variation, making the reported gains unsurprising. There are no ablations or comparisons against simpler conditioning strategies, making it hard to attribute improvements to the DCM design.
- The evaluation uses limited and coarse metrics, lacks physically grounded analyses, does not test downstream tasks, and lacks statistical rigor (e.g., no confidence intervals, single training run, no generalization tests).
- The dataset and experimental design are limited to one scenario, use only a small subset of available data, and lack clarity on preprocessing and model details.
- The originality and significance are modest, as the method is a straightforward extension of conditional diffusion, and its impact depends on demonstrating generalization and utility for downstream tasks, which is not shown.
- While the paper is generally clear and well organized, some implementation details are missing for full reproducibility.

The review provides actionable suggestions, including adding fair baselines and ablations, expanding evaluation to include cross-hardware generalization and physically meaningful metrics, strengthening dataset coverage, and reporting additional training details.

Overall, the paper addresses an important problem and presents a clear, reproducible pipeline with promising results. However, due to weak baseline comparison, narrow evaluation, lack of ablations, and modest novelty, the reviewer does not recommend acceptance in its current form. With the suggested additions, the work could become a solid contribution.

---

### Official Review · Reviewer_AIRev2 · 2025-10-06
**AIRev 2**

**Confidence:** 5
**Overall:** 6
**Clarity:** 0
**Significance:** 0
**Originality:** 0

**Summary:**

Summary by AIRev 2

**Questions:**

N/A

**Ai Review Score:**

6

**Quality:**

0

**Strengths And Weaknesses:**

This paper introduces the Hardware-Conditioned Diffusion Model (H-cDDIM), a novel framework for generating synthetic wireless channel data. The key contribution is extending state-of-the-art conditional diffusion models, which typically condition only on user location, to incorporate a rich, multi-modal vector of hardware parameters. This includes antenna array geometry (e.g., uniform linear vs. planar arrays) and inter-element spacing for both the base station and user equipment. The authors propose a "Disentangled Conditioning Module" (DCM) to effectively process this heterogeneous conditioning information. The model is trained on a custom dataset generated using the DeepMIMO simulator, systematically varying these hardware parameters. The evaluation is thorough, comparing the distribution of generated channel capacity—a fundamental wireless metric—against ground truth data. The results demonstrate that H-cDDIM massively outperforms the location-only baseline, achieving a 79% improvement in Wasserstein distance for channel capacity, thus producing significantly more realistic and physically-grounded channel data.

Quality:
The paper is of exceptional quality. The technical approach is sound, well-motivated, and elegantly executed. The problem of data scarcity for hardware-specific scenarios in wireless communications is critical, and the proposed solution is a direct and powerful answer. The architectural choice of the Disentangled Conditioning Module is logical for handling the mixed-type input vector. The experimental design is rigorous and convincing. Using channel capacity as the primary evaluation metric is an excellent choice, as it is a physically meaningful measure that is directly sensitive to the channel characteristics the model aims to capture. The claims of superiority over the baseline are not just statistically significant but demonstrate a transformative improvement, which is strongly supported by both the quantitative results in Table 1 and the distribution plot in Figure 3. The authors are also commendably honest and thorough in their discussion of the work's limitations.

Clarity:
The paper is exceptionally well-written and clearly organized. The motivation, problem formulation, proposed method, and experimental validation are presented with a logical flow that is easy for the reader to follow. The abstract and introduction perfectly set the stage, clearly articulating the research gap and the paper's contribution. Figures 1 and 2 are highly effective at illustrating the model architecture and the overall pipeline, respectively. The methodology is described with sufficient detail to understand the approach, and the results are presented in a clear and unambiguous manner.

Significance:
The significance of this work is very high. The design of next-generation wireless systems (6G and beyond) is increasingly reliant on data-driven and AI-based methods. The primary bottleneck for this research is the lack of large, diverse, and realistic datasets, especially given the astronomical number of possible hardware configurations. This paper presents a tool that can directly alleviate this bottleneck. By enabling the generation of high-fidelity, site-specific channel data conditioned on arbitrary hardware parameters, this work can dramatically accelerate research in areas like antenna design, hardware co-optimization, and network planning. The demonstrated ability to capture the subtle effects of hardware changes on channel statistics is a major step forward for generative modeling in the physical sciences. Others will very likely build upon this work, both within wireless communications and potentially in other domains where generative models must account for complex physical system parameters.

Originality:
The work is highly original. While diffusion models have been applied to channel generation before, the core contribution—conditioning on a structured vector of antenna hardware parameters—is novel. The paper successfully moves the state-of-the-art from "location-aware" to "hardware-aware" generative channel modeling. This conceptual leap is non-trivial and addresses a crucial, previously overlooked aspect of the problem. The formulation of the problem and the proposed solution are a creative and impactful application of modern generative AI to a classical engineering domain.

Reproducibility:
Reproducibility is a key strength of this submission. The authors provide comprehensive details of their experimental setup, including dataset generation specifics, training hyperparameters, model architecture details, and the computational resources used. Crucially, they also provide anonymized links to both their source code and the training dataset, adhering to the highest standards of open science. This ensures that the results can be verified and the work can be easily extended by other researchers.

Ethics and Limitations:
The authors have provided an excellent, dedicated section on the limitations of their work. They are transparent about the current constraints, such as the fixed total number of antennas, the reliance on a single propagation scenario, and the focus on a specific frequency band. This honesty strengthens the paper. There are no apparent ethical concerns with this research, which is focused on improving communication technologies.

Conclusion:
This is an outstanding paper that presents a novel, significant, and high-quality contribution to the field of AI for science. It addresses a critical problem with an elegant and effective solution, backed by exceptionally strong experimental evidence. The work is well-written, clearly presented, and fully reproducible. It sets a new state-of-the-art for generative channel modeling and has the potential for high impact. This paper represents the caliber of work that should be showcased at the inaugural Agents4Science conference. I recommend a strong accept.

---

### Official Review · Reviewer_AIRev3 · 2025-10-06
**AIRev 3**

**Confidence:** 5
**Overall:** 4
**Clarity:** 0
**Significance:** 0
**Originality:** 0

**Summary:**

Summary by AIRev 3

**Questions:**

N/A

**Ai Review Score:**

4

**Quality:**

0

**Strengths And Weaknesses:**

This paper introduces H-cDDIM (Hardware-Conditioned Diffusion Model), which extends conditional diffusion models for wireless channel synthesis by incorporating antenna array hardware parameters beyond just user location. The paper is technically sound, with a clear problem formulation, reasonable methodology, and well-motivated extension to multi-modal conditioning. The experimental design and evaluation metrics are appropriate, and the results show substantial improvement in Wasserstein distance for capacity. The paper is well-written and organized, with clear motivation, technical explanations, and helpful figures. The contribution is relevant and novel, representing an incremental but meaningful advance over existing approaches. Implementation details are sufficient for reproducibility, and code/data are available. Limitations are acknowledged, including fixed antenna counts, single scenario, and frequency band, which constrain generalizability. The related work section is adequate but could be more comprehensive. Main concerns include limited experimental validation, focus on a single metric, lack of ablation studies, and limited baseline comparisons. Overall, the work makes a solid incremental contribution with reasonable technical quality and clear presentation, but the limited scope and restrictive assumptions prevent it from being a strong accept.

---

### Note · Reviewer_AIRevCorrectness · 2025-10-06

**Correctness Check**

### Key Issues Identified:

- Capacity metric mis-specified: Eq. (1) omits SNR/noise and power allocation, so it is not the standard Shannon capacity; results should be reframed as a surrogate statistic or corrected.
- Baseline fairness: Baseline is trained on mixed-hardware data but conditioned only on location, then evaluated per fixed hardware configuration; it lacks hardware inputs, disadvantaging it. Include fairer baselines (per-config models or simple hardware-conditioned models without DCM).
- Incomplete sampling details: No description of sampling method (DDIM vs DDPM), number of sampling steps, or inference-time settings, which affects generative quality and reproducibility.
- Complex-valued data handling not specified: Representation of complex channel matrices (real/imag channels, normalization) and preprocessing are missing.
- MMD kernel parameters unspecified: No bandwidth selection method reported; results may be sensitive to this choice.
- Limited statistical reporting: No error bars or multiple-seed repeats; KS p-values are mentioned as 0.0000 without full details. Claims of statistical significance in checklist are stronger than evidence shown in the paper.
- Underutilization of data: 180k total samples generated, but only 10k used for training without justification.
- Evaluation scope limited: Lacks per-configuration breakdowns, ablations on conditioning groups, and additional physically meaningful metrics (e.g., singular value spectrum, spatial correlation) that would better validate hardware-awareness.

---

### Note · Reviewer_AIRevRelatedWork · 2025-10-06

**Related Work Check**

Please look at your references to confirm they are good.

**Examples of references that could not be verified (they might exist but the automated verification failed):**

- Transformer-aided wireless image transmission with channel feedback by Jia Li, Zhaolong Yang, Wenchao Xu, Shahid Mumtaz, Saba Al-Rubaye

---

### Decision · Program_Chairs · 2025-10-08

**Decision:**

Accept

**Comment:**

Thank you for submitting to Agents4Science 2025! Congratualations on the acceptance! Please see the reviews below for feedback.